# Social, psychological and health characteristics associated with stability and change in adult alcohol consumption

Martin Ekholm Michelsen[1], Marie Grønkjær[1,2,3], Erik Lykke Mortensen[1,2], Cathrine Lawaetz Wimmelmann[1,2]*

1 Unit of Medical Psychology, Section of Environmental Health, Department of Public Health, University of Copenhagen, Copenhagen, Denmark, 2 Center for Healthy Aging, University of Copenhagen, Copenhagen, Denmark, 3 Center for Clinical Research and Prevention, Bispebjerg and Frederiksberg Hospital, Frederiksberg, Denmark

* calw@sund.ku.dk

## Abstract

### Background

Many studies analyzing health effects of alcohol consumption have operationalized alcohol intake from a single baseline measure without further follow-up. Consequently, there is a lack of knowledge about stability and change in alcohol consumption over the life course and the social, psychological, lifestyle, and health characteristics associated with different alcohol consumption trajectories.

### Objectives

The aims of the study were to describe the prevalence of different adult-life alcohol consumption trajectories among Danish men and to analyze social, psychological, lifestyle and health characteristics associated with these trajectories.

### Methods

For 2510 Danish men, retrospective decade-based information on alcohol consumption during life period 26–60 years was obtained in late midlife and information on individual characteristics was obtained in young adulthood, late midlife and from national hospital registries. The men were allocated to one of six a priori defined alcohol consumption trajectories.

### Results

About 65% of Danish men had a stable moderate consumption, drinking 1–21 units weekly while the five other consumption trajectories were comparatively rare: 3% stable abstainers, 4.7% stable high-risk drinkers, 10.9% with increasing and 12.7% with decreasing consumption. Moderate consumption over the adult life-course was associated with the most favorable social, psychological, lifestyle and health characteristics while the other trajectories were generally associated with less favorable characteristics to varying degrees–e. g. this

**Data Availability Statement:** The study includes data of a sensitive nature which, due to the low frequencies of some patterns of information, could compromise participant privacy even when

anonymized. The study was approved by the Danish Data Protection Agency, and in accordance with the Act on Processing of Personal Data (Act No. 429 of 31 May 2000) of the Danish Data Protection Agency, data thus cannot be made publicly available due to considerations for privacy and anonymity of the participants. However, an anonymized version of the full data set can be available to researchers who are qualified to handle confidential information in accordance with the Danish Data Protection Agency act. Contact Department of Public Health, Faculty of Health and Medical Sciences University of Copenhagen, Øster Farimagsgade 5, 1353 København K (https://ifsv.ku.dk; phone +45 35 32 76 23).

**Funding:** The author(s) received no specific funding for this work.

**Competing interests:** The authors have declared that no competing interests exist.

was the case for the stable abstaining trajectory and in particular the trajectory with decreasing consumption.

## Conclusion

The findings suggest that the majority of Danish men drink moderately in the life period from young adulthood to late midlife, and deviance from this 'normal' moderate consumption trajectory is associated with less favorable social, psychological, lifestyle and health characteristics. Some of these characteristics may influence alcohol consumption patterns, but for some of the trajectories, alcohol consumption may influence health as well as social and psychological functioning.

## Introduction

Alcohol is a major contributor to the global burden of disease. Globally, excessive alcohol consumption is related to 5.3% of all mortality while alcohol is the cause of the loss of 5.1% of all disability adjusted life years according to WHO global status report [1]. Although alcohol is a leading risk factor for death and disability, its overall association with health is complex as several observational studies have observed the U- or J-shaped curve suggesting that light to moderate drinking can have beneficial health outcomes [2, 3]. Although this may be true in some cases, many other biological and environmental factors related to alcohol consumption and health may also differ among abstainers from alcohol, moderate drinkers, and high-risk drinkers. Such factors may include sex, age, social factors, psychological characteristics, lifestyle as well as physical and mental health status. Furthermore, most studies analyzing associations of alcohol consumption with health and other characteristics have operationalized alcohol intake according to a single baseline measure without further follow-up [4]. Consequently, the stability of alcohol consumption over the life course has been a concern, in particular in discussions of characteristics of people who report abstaining from alcohol because information on drinking habits in earlier life periods is missing [5–7]. In fact, it has been suggested that abstinence from alcohol may be a consequence of alcohol-related health problems [6], even though numerous other factors could influence changes in alcohol habits during the life course. Thus, a British study based on data from nine cohorts observed increasing mean consumption in adolescence, a relatively stable midlife level and decline around age 60 [8]. However, even after age 60 both decrease and increase in consumption can be observed [9].

Alcohol consumption habits develop during adolescence and young adulthood and are less stable in this life period. This was confirmed in a meta-analysis finding dramatic shifts in alcohol intake in adolescence, but only moderate changes later in life [10], suggesting a shift from irregular drinking and binge-drinking in adolescence to a more stable drinking pattern later in adulthood. More recent studies have observed different trajectories of consumption in adolescence, including patterns of increasing and decreasing intake [11]. These findings and an observed peak around age 25 [8] suggest that a distinction should be made between adolescence and young adulthood where alcohol habits are being established and later adulthood where changes in consumption level and pattern can be observed but may be less frequent. Consequently, the present study focuses on adult trajectories of consumption after adolescence and before age-related changes in consumption after age 60 [9].

Our study sample consists of men only, and this raises the issue of sex differences in trajectories of alcohol consumption. Although men have higher levels of alcohol consumption than

women in most countries, including Denmark [12, 13], available evidence suggests that men and women have similar consumption trajectories across the life span [8]. However, there are relatively few relevant studies and more research comparing consumption trajectories in women and men is needed.

Some studies have focused on identifying consumption trajectories by analyses of empirical data. Thus, studies have used latent growth mixture modeling to identify different trajectories [11, 14] or used linear growth curve models to describe stability and change in consumption [4]. In contrast, we have defined consumption trajectories a priori to correspond to basic questions about adult alcohol consumption over the life course: What is the prevalence of abstaining during adult life, how prevalent and stable are moderate and high-risk consumption? Is increasing and decreasing consumption during adulthood common and how is change in consumption associated with social, psychological, lifestyle, and health characteristics? The aims of this explorative study are to describe the prevalence of different adult-life alcohol consumption trajectories among Danish men and to analyze social, psychological, lifestyle and health characteristics associated with these adult-life consumption trajectories. We expect that a trajectory reflecting moderate alcohol consumption during adult life will be the most prevalent among Danish men and that this trajectory will be associated with more favorable characteristics than other consumption trajectories. The latter assumption is corroborated by Danish studies showing that differences in alcohol consumption is associated with not only health-related characteristics, but also with a wide range of socio-demographic and psychological characteristics [12, 15, 16].

## Materials and methods

### Study design and participants

The study is based on participants from the Lifestyle and Cognition Follow-up study (LiKO-15) [17]. The men invited to LiKO-15 were selected from two existing databases [18, 19]. The sample comprises 2611 Danish men born between 1950 and 1961, draft board examined in 1968–89, and re-examined in 2015–17 [17]. The present study sample comprises 2510 men with draft board information as well as sufficient LiKO-15 follow-up information on adult-life alcohol consumption trajectories.

Information on alcohol consumption and individual characteristics was obtained from Draft Board registries (young adulthood) and LiKO-15 (late midlife). Additionally, hospital admission diagnoses were obtained from the Danish National Patient Registry and the Danish Psychiatric Central Research Register.

### Measures

**Adult-life alcohol consumption trajectories.** At the LiKO-15 follow-up, the participants completed a computer-based questionnaire, which included self-reported information on the weekly number of units over the entire life course, including the age periods 26–30 years, 31–40 years, 41–50 years, and 51–60 years. Data were also obtained for age-periods below age 26, but to focus on relatively stable adult drinking patterns, the analysis was restricted to the age-period 26–60 years. In Denmark, a standard drink or unit corresponds to 12 g of pure alcohol, roughly equivalent to 1 beer, 1 glass of wine or 4 cl of 40% liquor. The weekly number of units was calculated based on self-reported information on typical consumption of beer, wine, and liquor. Since the focus was on adult drinking trajectories, six a priori consumption trajectories were derived from the weekly number of units in the age periods 26–30 years, 31–40 years, 41–50 years, and 51–60 years. Men belonging to the first three stable trajectories were first identified, and then the remaining men were allocated to one of the other three trajectories:

*Abstaining*: This trajectory included participants who throughout the age period from 26 to 60 consistently reported drinking less than 1 unit per week.

*Moderate consumption*: This trajectory included men who throughout the age period from 26 to 60 consistently reported drinking between 1 and 21 units per week.

*High-risk consumption*: This trajectory included men who throughout the age period from 26 to 60 consistently reported drinking more than 21 units per week.

*Increasing consumption*: This trajectory included men who reported higher weekly alcohol consumption in the age period 51–60 years than in the age period 26–30 years.

*Decreasing consumption*: This trajectory included men who reported lower weekly alcohol consumption in the age period 51–60 years than in the age period 26–30 years.

*Fluctuating consumption*: This trajectory included participants who reported consuming the same amount of alcohol in the age periods 26–30 years and 51–60 years but deviated in the interim periods.

**Young adult data.** All Danish men–except those with disqualifying disease such as intellectual disability, diabetes, and epilepsy–must appear before the draft board at age 18 or seek permission to appear later. The draft board assessment includes a health examination and administration of a test of intelligence. Computerized records include weight, height, information on education and the intelligence score. For most of the sample (n = 2170), education was coded on a 1–9-point scale with 1 corresponding to basic school education without formal exam and 9 to university education (a less detailed code was used for the remaining men). The intelligence test, Børge Priens Prøve (BPP), is a 45-minute group-administered test of intelligence, consisting of four subtests assessing verbal reasoning and fluid intelligence [20]. From the four subtests a standardized IQ score with a mean of 100 and a standard deviation (SD) of 15 in the study sample was derived. The participating men appeared before the draft board at a mean age of 20.29 (SD = 2.05) and at that time, the BPP was administered as a paper-and-pencil test.

**Late midlife data.** The late midlife follow-up consisted of a computerized version of the BPP intelligence test and a computerized questionnaire, which in addition to alcohol consumption included questions on socio-demographic factors, personality, lifestyle, and self-reported health. The mean age at follow-up was 61.62 years (SD = 3.27) and the average time since appearing before the draft board was 41.3 years (SD = 3.35).

*Sociodemographic factors*. A variable reflecting number of years of education was generated based on a combination of self-reported information on school education and post-school vocational training and education. Based on relevant questions, binary variables were derived indicating having a job, having a spouse or partner, and having children.

*Intelligence*. At the late midlife follow-up, the BPP was re-administered in a computerized version. Using the sample Draft Board BPP mean and SD, the follow-up BPP score was also transformed to a standardized IQ scale. Thus, the difference between the follow-up and the original draft board BPP score could be used as measure of age-related change in intelligence.

*Personality*. A 10-item short version of the Big Five Inventory (BFI-10) was integrated in the computerized questionnaire [21]. The 10 items are presented in a 1–5 Likert format, and two items are included for each of the Big Five personality dimensions (neuroticism, extraversion, openness, agreeableness, and conscientiousness). For each personality dimension, we calculated the mean of the two relevant items and transformed the scores to z-scores with a sample mean of 0 and a SD of 1. The BFI-10 has been validated against the BFI-44 and the NEO-PI-R [21].

*Lifestyle*. For smoking, total pack years was derived based on the question 'How many cigarettes did you smoke per day (on the days you smoke) in each age period <15, 15–18, 19–25, 26–30, 31–40, 41–50, 51–60, and >60. Number of pack years was calculated among ever smokers using the following tobacco equivalents: 1 cigarette = 1 unit, 1 cigarillo = 3 units, 1 cigar = 5 units and 1 pipe = 3 units. A question was also asked about extreme binge drinking (more than 10 units on one occasion) during the age periods from 26 to 60. Based on this information, the number of years with weekly extreme binge drinking was counted. Moreover, questions were asked about the weekly number of hours spent on physical activity on a 1–7-point scale and on the intensity of physical activity on a 1–4-point scale. These two questions were submitted to a principal component analysis and as the first principal component explained 73% of the variance, it was used as a measure of physical activity with a mean of 0 and a SD of 1. Height and weight were self-reported at the follow-up examination. Body mass index (BMI) was calculated and compared with the draft board BMI to evaluate change in BMI.

*Self-reported health*. Physical and mental health was self-reported on a five-point Likert scale, but transformed to binary measures of good vs. bad physical and mental health, respectively. The Major Depression Inventory (MDI) comprising 10 items in a 0–5 point Likert format was used to assess self-reported depression with a total score range from 0 to 50 [22]. Several questions were also asked about current and previous health conditions and diseases. These questions were coded in a binary yes/no format: High cholesterol, high blood pressure, diabetes, heart attack, stroke, and concussion. Finally, a self-reported question on previous treatment for alcohol problems was also coded in a yes/no format.

**Hospital diagnoses.** Information on hospital admission diagnoses from the Danish National Patient Registry was used to calculate the Charlson Comorbidity Index [23]. The index includes 19 different comorbidities that are weighted by their potential influence on the risk of death. In the present study sample, the number of comorbidities had a range from 0 to 16.

Information on hospital diagnoses from psychiatric wards obtained from the National Patient Registry and the Psychiatric Central Research Registry was used to assess mental disorders over the life course. ICD-8 and ICD-10 diagnoses were derived from the registry information that was updated in January 2018 when the mean age of the participants was 62.62 (SD = 3.22). The following ICD-8 and ICD-10 diagnoses were registered:

1. Alcohol-related disorders: 303.x diagnoses and F10.x diagnoses.

2. Substance Abuse: 304.x and F11.x –F19.x diagnoses.

3. Psychotic disorders: 295.x, 297.x, 298.x diagnoses and F20.x diagnoses.

4. Mood disorders: 296.x diagnoses and F30–F39 diagnoses.

5. Neuroses/Anxiety disorders: 300.x diagnoses, F40.x–F42.x and F44.x–F49.x diagnoses.

6. Adjustment disorders: 307.x diagnoses and F43.x diagnoses.

7. Personality disorders: 301.x diagnoses and F60.x–F61.x or F69.x diagnoses.

## Data analysis

Differences among the six consumption trajectories were analyzed using regression models that in addition to dummy variables coding the consumption trajectories included adjustment for either birthdate or follow-up age. Late midlife follow-up data were analyzed in regression models adjusting for age at follow up while young adulthood data and health registry data

were analyzed in models adjusting for date of birth. The moderate consumption trajectory (1–21 weekly units) was used as reference in the analyses. Continuous variables were analyzed using linear regression, binary variables using logistic regression and count variables using zero-inflated negative binomial regression. All analyses were conducted with Stata version 17 and using Stata's vce option to obtain robust standard errors.

For most variables, there were either no missing data or less than 1 percent was missing. However, due to a technical error, data were missing for 45 men on the personality variables.

While the draft board data generally can be assumed to be unaffected by adult-life alcohol consumption, this is not the case for late midlife follow-up data and health registry data. Consequently, we have not conducted analyses that include the latter variables as covariates (e.g. adjusting for personality traits or psychiatric diagnoses).

## Ethics

According to current Danish law all health science projects must be reported to and approved by the local scientific ethics committee (§ 14 in the law). However, the law specifies several exceptions to this general rule, such as 14.2, which states that projects based on questionnaires and/or register data should only be reported and approved if the project also involves human biological material. Since the analyses do not involve biological material, this study does not require approval by the Danish scientific ethical committee system, but the study was approved by the Danish Data Protection Agency. When the participants were invited to participate, they received written information about the purpose of the study and the study procedures. Based on this information, they decided whether to participate or not.

## Results

The characteristics of the full sample are shown in the first column of Tables 1–4. The average weekly number of units in the full sample was relatively stable across adult-life age periods (range 11.6–12.9 units per week) (Table 1). Most men reported to have good physical (85.9%) and mental health (91.2%). The sample included 10.3% who reported that they had been in treatment for alcohol problems (Table 3), and 7.8% with alcohol-related hospital diagnoses (Table 4).

Table 1 shows the prevalence of the six adult-life consumption trajectories. Among the participants, 3.0% reported to abstain from alcohol from age 26 to 60 years, while the majority (65.1%) reported a trajectory of stable moderate consumption, defined as drinking 1–21 units per week. Stable high-risk drinkers comprised 4.7% of the sample while 10.9% reported

**Table 1. Average number of units per week for each age period in six adult-life trajectories of alcohol consumption.**

| | Full sample | Abstaining | Moderate | High-risk | Increasing | Decreasing | Fluctuating |
|---|---|---|---|---|---|---|---|
| | N = 2510 men | n = 76 | n = 1635 | n = 119 | n = 274 | n = 320 | n = 86 |
| | | 3.0% | 65.1% | 4.7% | 10.9% | 12.7% | 3.4% |
| **Age periods** | | | | | | | |
| **26–30** | 11.56 | 0.00 | 7.87 | 35.50 | 9.22 | 25.78 | 13.38 |
| **31–40** | 12.86 | 0.00 | 8.22 | 39.91 | 14.23 | 26.49 | 20.13 |
| **41–50** | 12.75 | 0.00 | 8.75 | 45.28 | 19.35 | 16.77 | 18.93 |
| **51–60** | 11.85 | 0.00 | 9.08 | 46.73 | 25.26 | 3.98 | 13.49 |
| **Mean (unweighted)[a]** | 12.23 | 0.00 | 8.48 | 41.86 | 17.02 | 18.25 | 16.48 |

[a] Across age periods, 26–60

**Table 2. Birthdate-adjusted analyses of differences in young adulthood (draft board data) among six adult-life trajectories of alcohol consumption.**

|  | Full sample n = 2510 men | Abstaining n = 76 | Moderate n = 1635 | High-risk n = 119 | Increasing n = 274 | Decreasing n = 320 | Fluctuating n = 86 | P-value[a] |
|---|---|---|---|---|---|---|---|---|
| **Draft Board Data (continuous)** |  |  |  |  |  |  |  |  |
| **Intelligence (IQ)** | 100.0 (15.0) | 95.81** | 100.99 | 100.19 | 99.43 | 96.61*** | 99.01 | <0.001 |
| **Education (1–9 scale)** | 6.07 (1.92) | 5.73 | 6.22 | 6.07 | 6.12 | 5.44*** | 5.67* | <0.001 |
| **Weight** | 69.04 (8.87) | 68.80 | 69.09 | 69.85 | 69.36 | 68.58 | 67.97 | 0.581 |
| **Height** | 179.48 (6.60) | 179.09 | 179.69 | 179.88 | 179.74 | 178.32** | 178.67 | 0.019 |
| **BMI** | 21.48 (3.14) | 21.44 | 21.38 | 21.63 | 21.75 | 21.76 | 21.26 | 0.489 |

[a] P-value for overall test of differences among the six trajectories

* P < 0.05

** P < 0.01

*** P < 0.001

increasing consumption during adult life and 12.7% reported decreasing consumption. Only 3.4% of the men reported a fluctuating trajectory characterized by periods with change in consumption patterns between young adulthood and late midlife.

Table 1 also shows the average weekly alcohol intake in the different adult-life periods for the six alcohol consumption trajectories. The overall mean for the moderate trajectory was 8.48 units per week with a range from 7.87 to 9.08 units per week, indicating a slight increase during adulthood. For the high-risk trajectory, the consumption means suggest a substantial increase during adulthood from 35.50 units per week in young adulthood to 46.73 units in late midlife. However, the distributions were quite skewed, and the corresponding medians suggest an increase from 30 units to 35 units. For the increasing trajectory, the mean consumption levels indicated moderate consumption in young adulthood (9.2 units per week), but gradually increasing to high-risk consumption in late midlife (25.26 units per week). For the decreasing trajectory, the pattern was reversed with high-risk consumption in the first two age periods and a very low consumption level in late midlife (3.98 units per week). Men with a fluctuating alcohol consumption trajectory showed relatively moderate consumption levels in both young adulthood and late midlife (about 13 units per week), but an increase in the age periods 31–40 and 41–50 years (about 19–20 units per week).

Table 2 shows young adult characteristics in relation to consumption trajectories. The highest intelligence score and educational level were observed among moderate drinkers, while participants who were abstaining from alcohol had a mean intelligence score of 95.81. The decreasing alcohol consumption trajectory was associated with significantly lower intelligence, educational level and height compared with moderate drinkers.

Table 3 presents late midlife characteristics in relation to consumption trajectories. Compared with moderate drinkers, participants who were abstaining or showed a decreasing consumption trajectory had significantly lower mean number of years of education. The percentages with a partner, children and with a job were highest among moderate drinkers and significantly lower among high-risk drinkers and participants with a decreasing consumption trajectory.

Compared with the moderate consumption trajectory the abstaining, the increasing and the decreasing consumption trajectories obtained significantly lower intelligence scores in late midlife (Table 3). The high-risk consumption trajectory showed the largest decline in intelligence scores, but there were no significant differences in decline among the six consumption trajectories.

**Table 3. Age-adjusted analyses of differences in late midlife (follow-up) among six adult-life trajectories of alcohol consumption.**

| | Full sample n = 2510 men | Abstaining n = 76 | Moderate n = 1635 | High-risk n = 119 | Increasing n = 274 | Decreasing n = 320 | Fluctuating n = 86 | P-value[a] |
|---|---|---|---|---|---|---|---|---|
| **Late Midlife Follow-up Data** | | | | | | | | |
| *Sociodemographic factors* | | | | | | | | |
| **Age at follow-up (continuous)** | 61.62 (3.27) | 61.60 | 61.63 | 62.37* | 62.03 | 61.13* | 61.11 | 0.001 |
| **Years of Education (continuous)** | 13.58 (2.52) | 13.03* | 13.79 | 13.36 | 13.61 | 12.73*** | 13.31 | <0.001 |
| **Job (binary)** | 65.30% | 0.59 | 0.69 | 0.49*** | 0.62* | 0.55*** | 0.63 | <0.001 |
| **Partner (binary)** | 81.79% | 0.76 | 0.85 | 0.68*** | 0.83 | 0.71*** | 0.73* | <0.001 |
| **Children (binary)** | 82.11% | 0.76 | 0.86 | 0.75** | 0.77** | 0.74*** | 0.77 | <0.001 |
| *Intelligence (continuous)* | | | | | | | | |
| **Intelligence (IQ)** | 94.91 (14.64) | 91.50* | 96.06 | 93.43 | 93.75* | 91.69*** | 93.73 | <0.001 |
| **IQ change** | -5.09 (9.33) | -4.33 | -4.93 | -6.75 | -5.69 | -4.94 | -5.29 | 0.490 |
| *Personality (continuous)* | | | | | | | | |
| **Neuroticism** | z-score | 0.08 | -0.04 | -0.01 | 0.07 | 0.10* | 0.05 | 0.222 |
| **Extraversion** | z-score | -0.23* | 0.03 | 0.12 | 0.02 | -0.14*** | -0.12 | 0.017 |
| **Openness** | z-score | -0.01 | -0.00 | 0.13 | -0.07 | 0.05 | -0.08 | 0.551 |
| **Agreeableness** | z-score | -0.10 | 0.04 | -0.03 | 0.05 | -0.16** | -0.16 | 0.026 |
| **Conscientiousness** | z-score | -0.07 | 0.05 | -0.04 | -0.01 | -0.19*** | -0.17* | 0.006 |
| *Lifestyle* | | | | | | | | |
| **Smoking: Pack Years (count)** | 19.51 (26.90) | 15.42 | 15.67 | 34.64*** | 21.89** | 29.58*** | 30.54*** | <0.001 |
| **Years with weekly Extreme Binge Drinking (count)** | 3.93 (9.49) | 0.06*** | 1.78 | 17.51*** | 5.79*** | 8.46*** | 6.51*** | <0.001 |
| **Physical activity (continuous)** | z-score | -0.30** | 0.08 | -0.16* | -0.10** | -0.15*** | -0.19* | <0.001 |
| **BMI (continuous)** | 26.59 (4.05) | 27.69** | 26.28 | 28.10*** | 26.83* | 27.19*** | 26.56 | <0.001 |
| **BMI change (continuous)** | 5.17 (3.67) | 6.27** | 4.91 | 6.35*** | 5.34 | 5.64** | 5.29 | <0.001 |
| *Self-reported Health (binary)* | | | | | | | | |
| **Good Physical health** | 85.98% | 0.82 | 0.90 | 0.76** | 0.83** | 0.76*** | 0.79* | <0.001 |
| **Good Mental health** | 91.21% | 0.87 | 0.93 | 0.84** | 0.88** | 0.87** | 0.89 | <0.001 |
| **Major Depression Inventory (continuous)** | 5.47 (6.34) | 6.99 | 4.73 | 7.91*** | 6.37*** | 6.93*** | 6.75** | <0.001 |
| **High Blood Pressure** | 37.22% | 0.30 | 0.36 | 0.46* | 0.38 | 0.41 | 0.44 | 0.088 |
| **High Cholesterol** | 32.19% | 0.26 | 0.30 | 0.39 | 0.36 | 0.35 | 0.37 | 0.083 |
| **Diabetes** | 8.79% | 0.12 | 0.07 | 0.18** | 0.08 | 0.13** | 0.17* | <0.001 |
| **Heart attack** | 5.67% | 0.12 | 0.05 | 0.04 | 0.05 | 0.09* | 0.06 | 0.108 |
| **Stroke** | 4.35% | 0.09 | 0.03 | 0.07 | 0.06 | 0.08** | 0.07 | 0.001 |
| **Concussion** | 35.84% | 0.30 | 0.34 | 0.45* | 0.32 | 0.45** | 0.37 | 0.002 |
| **Treatment for alcohol problems** | 10.32% | No data | 0.04 | 0.35*** | 0.10** | 0.32*** | 0.23*** | <0.001 |

[a] P-value for overall test of differences among the six trajectories

* P < 0.05

** P < 0.01

*** P < 0.001

Significant overall p-values on the personality traits of extraversion, agreeableness, and conscientiousness were found (Table 3). Compared with moderate drinkers, men with abstaining and decreasing consumption trajectories scored significantly lower on extraversion, and this was also the case for men with decreasing trajectory on agreeableness. Both the fluctuating and the decreasing trajectories were associated with significantly lower scores on conscientiousness compared with the moderate consumption trajectory. The overall p-values were not significant

**Table 4. Birthdate-adjusted analyses of differences in comorbidity index and prevalence of hospital admission diagnoses (registry data) among six adult-life trajectories of alcohol consumption.**

| | Full sample n = 2510 men | Abstaining n = 76 | Moderate n = 1635 | High-risk n = 119 | Increasing n = 274 | Decreasing n = 320 | Fluctuating n = 86 | P-value[a] |
|---|---|---|---|---|---|---|---|---|
| **Registry Data (binary)** | | | | | | | | |
| **Charlson Comorbidity Index (count)** | 0.87 (1.62) | 1.07 | 0.73 | 1.58** | 0.83 | 1.23*** | 1.33** | <0.001 |
| **Alcohol-related disorders** | 8.17% | 0.03 | 0.04 | 0.26*** | 0.07* | 0.25*** | 0.15** | <0.001 |
| **Substance abuse** | 2.11% | 0.05 | 0.01 | 0.03 | 0.02 | 0.07*** | 0.02 | <0.001 |
| **Psychotic disorders** | 4.70% | 0.10* | 0.03 | 0.03 | 0.06 | 0.08** | 0.10* | 0.001 |
| **Mood disorders** | 10.04% | 0.15 | 0.08 | 0.17* | 0.09 | 0.15** | 0.12 | 0.003 |
| **Neurosis/Anxiety disorders** | 6.45% | 0.16** | 0.05 | 0.09 | 0.08 | 0.09** | 0.11 | 0.002 |
| **Adjustment disorders** | 12.43% | 0.24** | 0.10 | 0.17* | 0.11 | 0.17** | 0.22* | <0.001 |
| **Personality disorders** | 8.76% | 0.20** | 0.06 | 0.14* | 0.10* | 0.17*** | 0.11 | <0.001 |

[a] P-value for overall test of differences among the six trajectories

* P < 0.05

** P < 0.01

*** P < 0.001

for neuroticism, but the decreasing consumption trajectory was associated with significantly higher scores than the moderate consumption trajectory.

The smoking level was similar for the abstaining and moderate consumption trajectories, while all the other consumption trajectories differed substantially from these two trajectories (Table 3). Higher average alcohol consumption was associated with larger number of years with weekly extreme binge drinking, and only the abstaining trajectory did not differ significantly from the moderate consumption trajectory. Moderate drinkers had the highest level of physical activity, and in fact, all other consumption trajectories were associated with significantly lower levels. The lowest level of physical activity was observed among participants with abstaining trajectory. The moderate drinkers had the lowest BMI and the smallest increase in BMI from young adulthood to late midlife, while the fluctuating trajectory was the only trajectory that did not deviate significantly from the moderate consumption trajectory. The largest increase in BMI was observed among participants with abstaining and high-risk consumption trajectories.

The moderate consumption trajectory was associated with the best self-reported physical and mental health, and most of the other consumption trajectories had significantly poorer self-reported health (Table 3). The scores on the Major Depression Inventory followed the trajectory of scores on self-reported health and showed a similar pattern of significant differences. On the remaining health variables, participants with abstaining and moderate consumption trajectories did not differ, while high blood pressure, diabetes, and concussions were significantly more prevalent among high-risk drinkers. For diabetes, this was also the case for the decreasing and the fluctuating consumption trajectories and for concussions the decreasing consumption trajectory only. Compared with moderate drinkers, heart attack and stroke were also more prevalent among participants with a decreasing alcohol consumption trajectory. Finally, apart from the abstaining trajectory, all other consumption trajectories were associated with higher prevalence of previous treatment for alcohol problems compared with the moderate drinkers. The highest prevalence was observed for high-risk drinkers and the decreasing consumption trajectories, while the lowest prevalence was observed for participants with increasing alcohol consumption trajectory.

Table 4 shows the prevalence of hospital diagnoses in relation to consumption trajectories. Moderate drinkers had the lowest score on the Charlson Comorbidity Index and lowest prevalence of most mental disorders. Compared with moderate drinkers, significantly higher scores on the Charlson Comorbidity Index (indicating worse health) and higher prevalence of alcohol-related diagnoses were observed in participants with high-risk, decreasing and fluctuating consumption trajectories. The prevalence of substance abuse diagnoses was about 2%, and only the decreasing consumption trajectory differed significantly from the moderate trajectory with respect to this diagnosis. Compared with moderate drinkers, significantly higher prevalence of psychotic disorders was observed for the abstaining, decreasing and the fluctuating consumption trajectories, while the increasing trajectory was marginally significant. Significantly higher prevalence of mood disorder was observed for the high-risk and decreasing consumption trajectories, and this was also the case for the abstaining and the decreasing trajectories with respect to neurosis or anxiety disorders.

Adjustment disorders were quite common (12.4%) and compared with the moderate consumption trajectory, the prevalence was significantly higher for the abstaining, high-risk, decreasing and fluctuating consumption trajectories. Almost 9% of the men were diagnosed with personality disorders and compared with the moderate consumption trajectory, the prevalence of personality disorders was significantly higher for the abstaining, high-risk, increasing and decreasing trajectories.

## Discussion

Most of the study sample belonged to the moderate consumption trajectory while the five other consumption trajectories were comparatively rare. The abstaining trajectory included the smallest number of men (n = 76), but this trajectory is important in relation to the large literature on abstaining, alcohol, and health. The analyses of sociodemographic factors, psychological characteristics, lifestyle, and health revealed several significant differences among men in the different consumption trajectories.

### Moderate consumption trajectory

About two out of three men showed a stable trajectory of moderate drinking within 1–21 units per week. This "normal" consumption trajectory was associated with favorable characteristics on most of the investigated variables. Thus, stable moderate alcohol consumption was associated with better social adjustment, desirable mental characteristics, healthy lifestyle, and good health. Even though the range of weekly units was quite broad for the moderate trajectory, the large number of men with this trajectory indicates considerable stability in alcohol habits for the majority of the study sample, and this is in line with similar British findings [4]. The average consumption levels in Table 1 also suggest substantial stability.

### Abstaining trajectory

The relatively few stable abstainers during adulthood deviate negatively on several social, psychological, lifestyle and mental health characteristics. Among the 76 abstainers, 31 individuals reported weekly moderate consumption before age 26. However, none of them reported high-risk drinking, and consequently, alcohol consumption in adolescence is unlikely to explain the associations with health. These findings are important since it has been suggested that negative deviation for abstainers primarily reflects the inclusion of former drinkers in the abstainer category [6, 24, 25]. In line with our findings, a previous study of young adult Danes observed low education, low intelligence, and personality deviance in abstainers [15] while a broad

range of cardiovascular risk factors have shown to be more prevalent among nondrinkers than among drinkers after adjustment for age and sex [26].

### High-risk consumption trajectory

This trajectory was observed in 119 men, indicating stable consumption of more than 21 units per week over the adult life span. An increasing and very high average level of consumption in late midlife was observed. However, only 72 (60.5%) men had higher consumption in late midlife (about 58 units) than in young adulthood (about 36 units per week) while the remaining 47 men showed a decrease in average consumption of about 4 units from young adulthood to late midlife. For the high-risk trajectory, it may be particularly informative to compare draft board data and late midlife follow-up data because the high consumption level may have influenced functioning and health over the life course. Compared with the moderate drinkers, the high-risk drinkers did not differ significantly with respect to young adult intelligence (in contrast to a previous Danish study [15]) and BMI, but in late midlife they showed the largest decrease in intelligence score and the largest increase in BMI of all consumption trajectories. The decrease in intelligence did not differ significantly from moderate drinkers, but a previous study using a different methodology observed significantly more decline in intelligence for adult-life consumption of more than 28 units per week [17]. The high-risk consumption trajectory was associated with the highest prevalence of several health problems, which may not only reflect the consistently high level of alcohol consumption, but also the high levels of extreme binge drinking and smoking (in line with [27]) as well as a generally unhealthy lifestyle [28]. The social functioning of high-risk drinking men has perhaps also been influenced by the high level of alcohol consumption as a significantly smaller percentage had a job, a partner, and children. A previous study of young adult Danes observed that high-risk drinking was associated with higher scores on neuroticism and a number of other personality traits [15], and a recent Danish study found that neuroticism prospectively predicts alcohol and substance abuse [29]. In line with these results, we observed significantly higher prevalence of adjustment and personality disorders among men with high-risk consumption trajectory.

### Increasing consumption trajectory

The increasing consumption trajectory was characterized by moderate drinking in young adulthood, but on average increasing about five weekly units for each age decade and reaching a high-risk average in the life-period 51–60 years. Participants with this trajectory were comparable on many characteristics to participants with a moderate consumption trajectory. Increased statistical power due to the relatively large group of men with this trajectory may partly explain some of the statistically significant differences. This seems to be the case for the significantly lower intelligence score and higher BMI at the late midlife follow-up, while the higher levels of smoking, extreme binge drinking, treatment for alcohol problems and alcohol-related diagnoses may reflect the drinking trajectory. The increasing trajectory also differed significantly from the moderate trajectory regarding self-reported health, self-reported depression and personality disorders. However, on several health variables, the men in the increasing consumption trajectory showed the second most favorable health status, only surpassed by participants with the moderate consumption trajectory, suggesting that the ability to increase alcohol consumption over the life span may require relatively good physical health.

### Decreasing consumption trajectory

The decreasing consumption trajectory was associated with low intelligence and educational level both in young adulthood and in late midlife. Moreover, suboptimal social functioning,

personality deviance and a less healthy lifestyle were observed in late midlife. Generally, the decreasing consumption trajectory was associated with remarkably poor physical and mental health characteristics, and this was confirmed by higher prevalence of all categories of mental disorder diagnoses. Particularly important was the high prevalence of alcohol-related diagnoses comparable to the prevalence for the high-risk trajectory as this suggests that the decreasing consumption trajectory includes a substantial number of men with previous high-risk consumption. This was also supported by the high mean consumption levels in the age period 26–40 years and the remarkably low consumption level in the age period 51–60. In fact, among the men with decreasing consumption trajectory, 70% reported abstaining from alcohol in the latter age period. Thus, a study defining abstainers by self-report in late midlife will tend to mix the relatively few life-long abstainers with a larger number of former high-risk drinkers, confounding social, psychological, lifestyle and health characteristics associated with abstaining [5, 6, 25]. A related issue is why previous high-risk drinkers are motivated and able to limit or abstain from alcohol intake in late midlife. Compared with the high-risk trajectory, the men with the decreasing trajectory seemed to have more health problems, and the substantial health problems may be a crucial factor both directly and indirectly influencing the experience and health consequences of alcohol consumption [9, 30]. Finally, low intelligence and educational level in young adulthood have been observed to predict both physical and mental health problems [15, 31, 32], and thus, the health problems associated with the decreasing consumption trajectory are not necessarily a consequence of excessive alcohol consumption in early life periods.

## Fluctuating consumption trajectory

This trajectory (comprising the second smallest part of the sample) is characterized by fluctuations in level of consumption over the adult life span even though men with this trajectory had the same level of consumption in young adulthood (26–30 years) and in late midlife (51–60 years). Among the 86 men, only six were abstainers in the two defining age periods, while eight men were high-risk drinkers in both periods, and closer analysis of consumption trajectories showed that most of the men had been high-risk drinkers at some point during the age period 31–50 years. Thus, the fluctuating trajectory includes a very heterogeneous group of men, and it may be difficult to interpret the observed associations. However, the trajectory was associated with the second highest smoking level, with several years of extreme binge drinking, and high levels of alcohol-related diagnoses and self-reported treatment for alcohol problems. The trajectory was also associated with significantly poorer physical and mental health. Most of the men had a decrease in consumption from age period 31–50 years to age period 51–60 years, and some of the findings are comparable to the results for the decreasing consumption trajectory.

## Strengths and limitations

There are several strengths in the present study, including the large study sample of men with detailed information on alcohol consumption trajectories and many different social, psychological, lifestyle and health characteristics, which both include self-reported information and information from national registries. The adult-life alcohol consumption information made it possible to describe different categories of abstainers and to analyze six groups with different consumption trajectories.

A major limitation is the fact that alcohol consumption during adult life was self-reported and therefore potentially inaccurate due to recall errors and bias. However, the associations between alcohol consumption trajectories and alcohol-related hospital diagnosis corroborate

the validity of the self-reported assessments. Moreover, the self-reported information was reliable enough to identify distinct social, psychological, lifestyle and health characteristics for six a priori defined consumption trajectories. Furthermore, the self-report method is supported by a previous British study using the retest method to evaluate the reliability of decade-based life-course self-report questions on alcohol consumption [33]. The study demonstrated not only reasonable reliability, but also correlations with prospective data on alcohol consumption.

Another limitation is the selection of the study sample. First, the LiKO-15 study only invited individuals living in a specific area in Copenhagen who had completed the BPP at draft board examinations, thus excluding men with disqualifying diseases and all women. Second, only 13% of the invited men participated in the study and these participants comprised a selected group with relatively high education and high intelligence in young adulthood [17]. Third, the invited population included a relatively large proportion of men with previous psychiatric admission and diagnosis, including alcohol-related hospital diagnoses [34]. This has obviously influenced the prevalence of mental disorders in the full sample and the prevalence of the six consumption trajectories as well as the social, psychological and health characteristics associated with the trajectories. Thus, it is likely that a more representative Danish sample would have a larger prevalence of both the abstaining and the moderate consumption trajectories as well as a lower prevalence of high-risk consumption. However, irrespective of the selected sample, the prevalence of the different consumption categories obviously reflects the Danish alcohol culture and may not be generalizable to other countries and cultures. In fact, some findings are dramatically different even when differences in sample and methodology are considered. Thus, a study of American military veterans observed about 65% to be abstinent or rare alcohol drinkers, while about 30% were moderate drinkers [14].

While our sample included a large number of men with a moderate consumption trajectory, the number of men with the other trajectories was relatively small, resulting in large standard errors and low statistical power. In particular, this was the case for the abstaining, the high-risk and the fluctuating trajectories, but in spite of potential power problems, we observed a number of significant differences between these trajectories and the moderate consumption trajectory.

Finally, we did not analyze the whole adult life span, but only the age period 26–60 years. Adolescence and young adulthood were excluded on purpose while the age of the participants prohibited analysis of consumption in old adulthood. However, average consumption seems to decline after age 60 [8], and changes in consumption in this life period may reflect both health status and social factors [9].

## Conclusions

The findings of this explorative study suggest that the majority of Danish men drink moderately in the life period from young adulthood to late midlife, and in a statistical sense stable moderate consumption is "normal" in Denmark. The study also suggests that moderate consumption over the adult life-course is associated with favorable social, psychological, lifestyle and health characteristics. Deviance from stable "normal" consumption includes trajectories of abstaining, high-risk consumption, increasing and decreasing consumption and generally, deviant trajectories are associated with less favorable characteristics to varying degrees depending on the alcohol consumption trajectory across the adult life span. Stable abstaining is rare in adulthood, but in late midlife, a relatively large number of former drinkers become abstainers. Decreasing consumption during adult life tends to be associated with substantial health problems as well as social and psychological deviance while this to a lesser extent is the case for men who increase consumption or are stable high-risk drinkers.

## Author Contributions

**Conceptualization:** Martin Ekholm Michelsen, Marie Grønkjær, Erik Lykke Mortensen, Cathrine Lawaetz Wimmelmann.

**Formal analysis:** Erik Lykke Mortensen.

**Investigation:** Martin Ekholm Michelsen, Marie Grønkjær, Erik Lykke Mortensen, Cathrine Lawaetz Wimmelmann.

**Methodology:** Martin Ekholm Michelsen, Marie Grønkjær, Erik Lykke Mortensen.

**Project administration:** Martin Ekholm Michelsen, Marie Grønkjær, Erik Lykke Mortensen, Cathrine Lawaetz Wimmelmann.

**Writing – original draft:** Martin Ekholm Michelsen.

**Writing – review & editing:** Martin Ekholm Michelsen, Marie Grønkjær, Erik Lykke Mortensen, Cathrine Lawaetz Wimmelmann.

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
