## [Decision Letter · Decision Letter 0]

22 Apr 2022

PONE-D-22-04437Social, psychological and health characteristics associated with stability and change in adult alcohol consumptionPLOS ONE

Dear Dr. Mortensen,

Thank you for submitting your manuscript to PLOS ONE. After careful consideration, we feel that it has merit but does not fully meet PLOS ONE’s publication criteria as it currently stands. Therefore, we invite you to submit a revised version of the manuscript that addresses the points raised during the review process.

We look forward to receiving your revised manuscript.

Kind regards,

Marc Potenza

Academic Editor

PLOS ONE

Journal Requirements:

Reviewers' comments:

Reviewer's Responses to Questions

**Comments to the Author**

1. Is the manuscript technically sound, and do the data support the conclusions?

Reviewer #1: Partly

Reviewer #2: Yes

2. Has the statistical analysis been performed appropriately and rigorously? 

Reviewer #1: Yes

Reviewer #2: Yes

3. Have the authors made all data underlying the findings in their manuscript fully available?

Reviewer #1: Yes

Reviewer #2: Yes

4. Is the manuscript presented in an intelligible fashion and written in standard English?

Reviewer #1: Yes

Reviewer #2: Yes

5. Review Comments to the Author

Reviewer #1: General Comments

Thank you for the opportunity to review this manuscript. The authors examined the prevalence and changes of alcohol consumption at two time points, as well as demographic and psychosocial correlates associated with alcohol use, among Danish men ages 26-60 years old. The current study has several strengths, including a large sample size and rich dataset. However, I have some concerns about the study rationale, analysis, and interpretation. I have provided several comments below, which I hope the authors will find helpful.

Major Comments

1. This study examines alcohol consumption over time among Danish men; however, the introduction currently indicates examining changes in alcohol use over time but does not specify why it should exclusively be studied among men. The authors should revise portions of the introduction to improve the framing of this study regarding their sample: Danish men. For example, the authors may wish to cite previous research that is more relevant to the current sample: Danish men or men in general.

2. In addition to the aims, what were the authors’ hypotheses for this study? There are many variables assessed in the current study (e.g., sociodemographic variables, IQ, personality, cigarette smoking, physical activity, BMI), and I am curious what the authors’ rationale was for including these variables. Please include hypotheses in the last paragraph of the introduction.

3. How were the drinking consumption trajectories derived? It is unclear from the method whether these were derived statistically (e.g. growth mixture modeling), based on prior literature and/or a priori hypotheses, or another way. The authors should elaborate on this point. After further reading, I see this is mentioned in the abstract, but not reported in the method or other main sections of the manuscript.

4. The sub-sample cell sizes for each drinking trajectory are quite variable; i.e., ranging from n=76 for abstainers to n=1,635 for moderate drinkers who represented over half of the total sample, as presented in Table 2. Given the authors’ findings of most favorable health outcomes associated with moderate drinkers who are also over-represented in this sample, is it possible that the large cell size drove these differences? In other words, what was done statistically to account for such different cell sizes?

5. Why were regression models used to analyze differences in consumption trajectories? In other words, what was the outcome/dependent variable across analyses? What was the referent category across analyses? It seems that a different type of analysis would be better suited to answer these research questions (e.g., ANOVA). If I am misinterpreting, addressing my second comment regarding indicating study hypotheses/research questions may also help to address the statistical tests chosen.

Minor Comments

6. The objectives should be changed from “adult-life” regarding alcohol consumption to “men” or “male” alcohol consumption to accurately reflect the current sample. Similarly, the focus on “adult” alcohol use mentioned should be changed to alcohol use among “men” or “males” throughout the manuscript.

7. What is the legal age of alcohol consumption in Denmark?

8. What was the exclusion criteria for the current study? Specifically, please talk more about what constituted “disqualifying disease.”

9. In the method, please clarify what “basic school education” indicates. The 1-9 point coding scale to measure education seems potentially problematic if 1 = basic school education. What about those who dropped out/did not complete school?

10. My next comments/questions involve the inclusion of an intelligence measure for this study, as assessed by the Børge Priens Prøve (BPP). Why was IQ assessed in this study? How is it relevant to alcohol consumption? What was the mean and SD of the IQ from the BPP among this sample? Lastly, it seems problematic to use changes in IQ score, as some research indicates that IQ is determined by a certain age and does not typically vary (excluding brain injuries and normative cognitive decline due to aging, for example). Assessing changes in academic performance or achievement over time would be a better indicator.

11. It seems problematic that the Major Depression Inventory, which was seemingly the only measure of mental health, was included as one of the key measures in the “self-reported health” section. I suggest adding depression as its own section with a header to clearly indicate it.

12. I recommend that the authors provide the 19 comorbidities assessed in the Charlson Comorbidity Index, or provide several examples.

13. The following sentence in the result section could be potentially harmful and should be tempered given that this mean (i.e., M = 95.81) still falls within the normal/average range for IQ, “…participants who were abstaining from alcohol had the lowest intelligence score.”

14. The authors should clarify the following sentence in the results section, “The scores on the Major Depression Inventory followed the trajectory of scores on self-reported health and showed a similar pattern of significant differences.” What does this mean for levels of depression among this sample?

Reviewer #2: I found the abstract well written and the sample size/scope impressive. I found the rationale in the introduction strong and easy to follow.

In the measures section, it was unclear to me whether the weekly number of units consumed over the entire life course was collected via timeline follow back or some other standardized tool. Perhaps it was similar to the cigarette pack years? Clarifying this would strengthen the paper. Moreover, it was unclear why the bottom age period was 26-30 instead of 21-30. I assume this was a function of the way the questionnaire was programmed. Finally, it was unclear why between 1 and 21 drinks per week was categorized as moderate consumption—this is a very large group with a wide range of consumption habits. More information here would be useful, world-count permitting. The rationale for transforming this measure to a binary measure of good vs. bad health was also unclear.

In the data analysis section, it wasn’t stated why moderate consumption trajectory (through presumably because this was the largest group) was used as the reference, but otherwise the analyses seemed reasonable. Some of the descriptive statistics in the beginning of the results are repetitive as these are already described in the prior sections. The discussion nicely highlights some key findings including that other studies “defining abstainers by self-report in late midlife will tend to mix the relatively few life-long abstainers with a larger number of former high-risk drinkers.”

My overall impressions of the manuscript are favorable.

6. PLOS authors have the option to publish the peer review history of their article (what does this mean?). If published, this will include your full peer review and any attached files.

Reviewer #1: No

Reviewer #2: No

---

## [Author Response · Author response to Decision Letter 0]

17 May 2022

PONE-D-22-04437

Social, psychological and health characteristics associated with stability and change in adult alcohol consumption

PLOS ONE

Author response to journal requirements and reviews

Journal Requirements:

Response: We have carefully checked that the manuscript meets the journal’s format and style requirements.

Response: We have revised the data availability statement to refer to the Department of Public Health, including full address and phone number. 

 The study includes data of a sensitive nature which, due to the low frequencies of some patterns of information, could compromise participant privacy even when anonymized. In accordance with the Act on Processing of Personal Data (Act No. 429 of 31 May 2000) of the Danish Data Protection Agency, data thus cannot be made publicly available due to considerations for privacy and anonymity of the participants. However, an anonymized version of the full data set can be available to researchers who are qualified to handle confidential information in accordance with the Danish Data Protection Agency act. Contact Department of Public Health, Faculty of Health and Medical Sciences University of Copenhagen, Øster Farimagsgade 5, 1353 København K (https://ifsv.ku.dk; phone +45 35 32 76 23)

Response: “Data not shown” has been a standard terminology to indicate a certain analysis is not presented in detail in a table. We used this terminology several places in the discussion and understand this has led to the misunderstanding that we describe another dataset, which is not available. Actually, we refer to specific analysis of our main database, which is available according to the data availability statement: 

“However, the distributions were quite skewed, and the corresponding medians suggest an increase from 30 units to 35 units (data not shown)”. (page 12 in the original manuscript).

“Among the 76 abstainers, 31 individuals reported weekly moderate consumption before age 26 (data not shown)”. (page 19 in the original manuscript).

“In fact, among the men with decreasing consumption trajectory, 70% reported abstaining from alcohol in the latter age period (data not shown)”. (page 21 in the original manuscript)

“Among the 86 men, only six were abstainers in the two defining age periods, while eight men were high-risk drinkers in both periods, and closer analysis of consumption trajectories showed that most of the men had been high-risk drinkers at some point during the age period 31-50 years (data not shown)”. (page 22 in the original manuscript)

“Most of the men had a decrease in consumption from age period 31-50 years to age period 51-60 years (data not shown)”. (page 22 in the original manuscript)

We believe that it is rather clear from the context that we in all cases refer to the database that is the basis for all analyses in the manuscript. However, since the terminology “data not shown” can cause confusion, we have deleted the statements from the manuscript. 

We find that this the “data not shown” statement is unnecessary in all cases and assume that the editor will agree. 

Reviewers' comments:

5. Review Comments to the Author

Reviewer #1: General Comments

Thank you for the opportunity to review this manuscript. The authors examined the prevalence and changes of alcohol consumption at two time points, as well as demographic and psychosocial correlates associated with alcohol use, among Danish men ages 26-60 years old. The current study has several strengths, including a large sample size and rich dataset. However, I have some concerns about the study rationale, analysis, and interpretation. I have provided several comments below, which I hope the authors will find helpful.

Major Comments

1. This study examines alcohol consumption over time among Danish men; however, the introduction currently indicates examining changes in alcohol use over time but does not specify why it should exclusively be studied among men. The authors should revise portions of the introduction to improve the framing of this study regarding their sample: Danish men. For example, the authors may wish to cite previous research that is more relevant to the current sample: Danish men or men in general.

Response: The reviewer rightly points out that this issue should have been discussed in the introduction. We don’t think that changes in alcohol use over time should exclusively be studied among men, but we focus on men because the available study sample only included men. In the strengths and limitations section the exclusion of all women is mentioned as a limitation, but we have added the following in a new paragraph in the introduction: 

“Our study sample consists of men only, and this raises the issue of sex differences in trajectories of alcohol consumption. Although men have higher levels of alcohol consumption than women in most countries, including Denmark [12, 13], available evidence suggests that men and women have similar consumption trajectories across the life span [8]. However, there are relatively few relevant studies and more research comparing consumption trajectories in women and men is needed”. (revised manuscript page 5, line 92).

2. In addition to the aims, what were the authors’ hypotheses for this study? There are many variables assessed in the current study (e.g., sociodemographic variables, IQ, personality, cigarette smoking, physical activity, BMI), and I am curious what the authors’ rationale was for including these variables. Please include hypotheses in the last paragraph of the introduction.

Response: Our study is essentially explorative, and this is the reason that we have not formulated hypotheses. However, to accommodate the reviewers concern, we have added the following to the paragraph describing the aims of the study:

“We expect that a trajectory reflecting moderate alcohol consumption during adult life will be the most prevalent among Danish men and that this trajectory will be associated with more favorable characteristics than other consumption trajectories. The latter assumption is corroborated by Danish studies showing that differences in alcohol consumption is associated not only with health-related characteristics, but also with a wide range of socio-demographic and psychological characteristics [12, 15, 16].” (revised manuscript page 6, line 109).

3. How were the drinking consumption trajectories derived? It is unclear from the method whether these were derived statistically (e.g. growth mixture modeling), based on prior literature and/or a priori hypotheses, or another way. The authors should elaborate on this point. After further reading, I see this is mentioned in the abstract, but not reported in the method or other main sections of the manuscript.

Response: It is true, that the method section does not explicitly use the term “a priori trajectories”, but the last paragraph in the introduction includes the following text “Some studies have focused on identifying consumption trajectories by analyses of empirical data. Thus, studies have used latent growth mixture modeling to identify different trajectories [11, 12] or used linear growth curve models to describe stability and change in consumption [4]. In contrast, we have defined consumption trajectories a priori to correspond to basic questions about adult alcohol consumption over the life course” (page 5, line 92 in the original manuscript). This is further elaborated in the method section “Since the focus was on adult drinking trajectories, six consumption trajectories were derived from the weekly number of units in the age periods 26–30 years, 31–40 years, 41–50 years, and 51–60 years. Men belonging to the first three stable trajectories were first identified, and then the remaining men were allocated to one of the other three trajectories” (original manuscript page 6, line 121). 

To accommodate the reviewer’s concerns, we have now inserted “a priori defined” in the method section “Since the focus was on adult drinking trajectories, six a priori defined consumption trajectories were derived from the weekly number of units in the age periods 26–30 years, 31–40 years, 41–50 years, and 51–60 years” (revised manuscript page 7, line 134).

4. The sub-sample cell sizes for each drinking trajectory are quite variable; i.e., ranging from n=76 for abstainers to n=1,635 for moderate drinkers who represented over half of the total sample, as presented in Table 2. Given the authors’ findings of most favorable health outcomes associated with moderate drinkers who are also over-represented in this sample, is it possible that the large cell size drove these differences? In other words, what was done statistically to account for such different cell sizes?

Response: For several reasons we compared the other 5 trajectories with the moderate consumption trajectory, which would have small standard errors and associated confidence intervals due to the large cell size. However, the significance of comparison with this trajectory would also depend on the cell size of each of the other 5 trajectories. Thus, on page 20, line 415 in the original manuscript we suggest that “Increased statistical power due to the relatively large group of men with this trajectory [the increasing consumption trajectory] may partly explain some of the statistically significant differences” and suggest that this may be the case for midlife intelligence and BMI. Table 3 shows that the intelligence level was 96.06, 93.75, and 93.73 for the moderate, increasing and fluctuating trajectories (with cell sizes of 1635, 274, and 86). Due to variance in cell size the standard errors of the estimated means were 0.35, 0.89, and 1.55 respectively, and the standard errors of the contrast comparing the increasing and fluctuating trajectory means with that of the moderate consumption trajectory were 0.95 and 1.59 respectively. Thus, the statistical power of these comparisons to a large extent depends on the cell size of the trajectory with the small cell size. We find this reasonable because the mean with the moderate trajectory are the most healthy and the cell size is large enough to serve as a normative reference.

To accommodate the reviewer’s concerns we discuss this problem in the strengths and limitations section of the revised manuscript:

“While our sample included a large number of men with a moderate consumption trajectory, the number of men with the other trajectories was relatively small, resulting in large standard errors and low statistical power. In particular, this was the case for the abstaining, the high-risk and the fluctuating trajectories, but in spite of potential power problems, we observed a number of significant differences between these trajectories and the moderate consumption trajectory”. (revised manuscript page 24, line 541).

5. Why were regression models used to analyze differences in consumption trajectories? In other words, what was the outcome/dependent variable across analyses? What was the referent category across analyses? It seems that a different type of analysis would be better suited to answer these research questions (e.g., ANOVA). If I am misinterpreting, addressing my second comment regarding indicating study hypotheses/research questions may also help to address the statistical tests chosen.

Response: The first paragraph in the data analysis section provides most of this information:

“Differences among the six consumption trajectories were analyzed using regression models.

Late midlife follow-up data were analyzed in regression models adjusting for age at follow

up while young adulthood data and health registry data were analyzed in models adjusting for

date of birth. The moderate consumption trajectory (1–21 weekly units) was used as

reference in the analyses. Continuous variables were analyzed using linear regression, binary

variables using logistic regression and count variables using zero-inflated negative binomial 0 regression.” (original manuscript page 19, line 214)

In other words, the characteristics listed in tables 2, 3, and 4 were outcome variables in regression analyses testing differences between the six trajectories and using the moderate consumption trajectory as reference. We could have used ANOVA to analyze the continuous variables, but this was not possible for binary and count variables. However, since the publication of Cohen’s book, behavioral scientists have been aware that ANOVA and multiple linear regression are both applications of the same general linear model (Cohen & Cohen, 1975 p. 186) and that results are identical for identical ANOVA and regression models. In our case, the models included a categorical variable (the 6 trajectories) and one continuous variable (either birthdate or age), and thus the regressions models correspond to analysis of covariance. STATA’s regression procedure was preferred over STATA’s ANOVA/ANCOVA procedure because it can provide robust standard errors. The use of robust standard errors was not mentioned in the original manuscript, but is now included in the slightly expanded data analysis paragraph:

“Differences among the six consumption trajectories were analyzed using regression models that in addition to dummy variables coding the consumption trajectories included adjustment for either birthdate or follow-up age. Late midlife follow-up data were analyzed in regression models adjusting for age at follow up while young adulthood data and health registry data were analyzed in models adjusting for date of birth. The moderate consumption trajectory (1–21 weekly drinks) was used as reference in the analyses. Continuous variables were analyzed using linear regression, binary variables using logistic regression and count variables using zero-inflated negative binomial 0 regression. All analyses were conducted with Stata version 17 and using Stata’s vce option to obtain robust standard errors” (revised manuscript page 11, line 226).

Minor Comments

6. The objectives should be changed from “adult-life” regarding alcohol consumption to “men” or “male” alcohol consumption to accurately reflect the current sample. Similarly, the focus on “adult” alcohol use mentioned should be changed to alcohol use among “men” or “males” throughout the manuscript.

Response: We understand the reviewer’s point, but also find it important to stress that we analyze consumption in adult life. Consequently, we have stressed “men” and “male” throughout the manuscript, but have also kept “adult life” where relevant.

7. What is the legal age of alcohol consumption in Denmark?

Response: There is no legal age of alcohol consumption in Denmark, but currently you need to be at least 16 years to be allowed to buy beer etc. (alcohol percentage below 16.5%) and 18 years to be allowed to buy liquor (alcohol percentage above 16.5%).

8. What was the exclusion criteria for the current study? Specifically, please talk more about what constituted “disqualifying disease.”

Disqualifying diseases are conditions that make men unfit for military service, such as mental retardation, diabetes or epilepsy. It is estimated that about 5-10% of the men born between 1950 and 1961 were exempted from appearing before a draft board (Gunhild et al., 2015).

We have provided examples of these diseases in the revised manuscript: “All Danish men – except those with disqualifying disease such as mental retardation, diabetes, and epilepsy – must appear before the draft board at age 18 or seek permission to appear later.” (Revised manuscript page 7, line 152)

9. In the method, please clarify what “basic school education” indicates. The 1-9 point coding scale to measure education seems potentially problematic if 1 = basic school education. What about those who dropped out/did not complete school?

Response: The 1-9 point scale used by the Danish military has the following steps:

1 = 7 years of school education without formal exam

2 = 7 years of school education supplemented by courses of general or practical education or 8 years of school education without formal exam

3 =8 years of school education supplemented by courses of general or practical education or ≥9 years of school education without formal exam

4 = trade apprenticeship

5 = 9 years of advanced school education with formal exam and diploma

6 = 10 years of advanced school education with formal exam and diploma, commercial or technical education, or high school without diploma

7 = non-academic professional training (e.g. teaching or engineering)

8 = high school formal exam and diploma

9 = university degree

This scale has been found very useful in research. In our sample it correlates 0.68 with BPP scores in young adulthood and 0.59 with middle age scores. However, due to changes in the Danish educational system, the Danish Military has introduced a more simple scale. This scale is much less informative, and this is the reason we decided to only include analyses of the 1-9 point scale.

Those who dropped out or did not complete school were scored 1 on this scale.

The original manuscript includes the following information:

“For most of the sample (n =2170), education was coded on a 1–9-point scale with 1 corresponding to basic school education and 9 to university education (a less detailed code was used for the remaining men)” (page..)

In the revised manuscript we have added “without formal exam” to the paragraph: 

“For most of the sample (n =2170), education was coded on a 1–9-point scale with 1 corresponding to basic school education without formal exam and 9 to university education (a less detailed code was used for the remaining men)” (revised manuscript page 8, line 156).

Since education is only one of many variables, we find that this description is sufficient, but should the editor disagree and prefer to include the detailed description of the 1-9 point scale, it can of course be included (or submitted as additional online material).

10. My next comments/questions involve the inclusion of an intelligence measure for this study, as assessed by the Børge Priens Prøve (BPP). Why was IQ assessed in this study? How is it relevant to alcohol consumption? What was the mean and SD of the IQ from the BPP among this sample? Lastly, it seems problematic to use changes in IQ score, as some research indicates that IQ is determined by a certain age and does not typically vary (excluding brain injuries and normative cognitive decline due to aging, for example). Assessing changes in academic performance or achievement over time would be a better indicator.

Response:

Draft board information was a requirement for the inclusion in the LIKO-15 sample, and consequently the BPP was available in our study sample. We report on the BPP because Danish studies have found associations of alcohol consumption and AUD diagnoses with low intelligence (Mortensen et al., 2006; Grønkjær et al,2019a; Mortensen et al, 2005; Grønkjær et al., 2019b).

The relevance to alcohol consumption is demonstrated in the references above.

The young adult mean and SD were by definition 100 and 15 (table 2), while the midlife mean and SD were 94.91 and 14.64 (table 3). Tables 2 and 3 shows the IQ mean for the 6 alcohol consumption trajectories and confirm associations between intelligence and alcohol consumption patterns.

Concerning the stability of intelligence: It is important to distinguish between stability of individual differences (rank-order consistency) and stability of mean level of performance. The correlation between young adult and midlife IQ is 0.80 in our sample and thus demonstrates considerable rank-order stability. However, this does not mean that the mean level of performance in IQ tests is constant across the life span. For the present sample, table 3 shows a mean decline of 5 IQ points corresponding to one third of a standard deviation. Furthermore, previous publications based on the LIKO-15 sample have demonstrated associations of lifetime alcohol consumption and alcohol diagnoses with decline on the BPP (Grønkjær et al,2019a; Grønkjær et al., 2019b), indicating that the consumption trajectories may be associated with cognitive decline. In our discussion of the high-risk consumption trajectory, we refer to the findings of the Grønkjær study (revised manuscript page 20, line 413).

11. It seems problematic that the Major Depression Inventory, which was seemingly the only measure of mental health, was included as one of the key measures in the “self-reported health” section. I suggest adding depression as its own section with a header to clearly indicate it.

Response: The MDI was the only multi-item measure of mental health, but a single item measure indicating the general level of mental health on a 1-5 point scale was also included (and so was a question on treatment for alcohol problems). However, we find the detailed information on diagnoses from admission to psychiatric hospital departments in table 4 perhaps more important than the self-report measures, and we don’t find it necessary to include a separate “mental health section” in table 3. Should the editor disagree, table 3 can of course be adjusted.

12. I recommend that the authors provide the 19 comorbidities assessed in the Charlson Comorbidity Index, or provide several examples.

Response: The comorbidity index of Charlson et al. (1987) includes the following diseases: Myocardial infarct, congestive heart failure, peripheral vascular disease, cerebrovascular disease, dementia, chronic pulmonary disease, connective tissue disease, ulcer disease, mild liver disease, diabetes, hemiplegia, moderate or severe renal disease, diabetes with end organ damage, any tumor, leukemia, lymphoma, moderate or severe liver disease, metastatic solid tumor, AIDS. This list can of course be included in the manuscript, but this comorbidity index is widely used, and we leave to the editor to decide whether it should be included (the index is rarely described in detail when used as a covariate).

13. The following sentence in the result section could be potentially harmful and should be tempered given that this mean (i.e., M = 95.81) still falls within the normal/average range for IQ, “…participants who were abstaining from alcohol had the lowest intelligence score.”

Response: The sentence that the reviewer is referring to only points to the fact that men with the abstaining trajectory had the lowest mean intelligence score in young adulthood. In other words, the sentence describes what the table presents. We are sure that the readers of Plos One will read this sentence appropriately: IQ is a continuous variable approximating a normal distribution with a mean of 100 an SD of 15, and a mean IQ of 95.81 implies that a substantial number of abstainers will in fact have an IQ above 100. Thus, the sentence in no way implies a direct relationship between abstaining and IQ below the normal range.

14. The authors should clarify the following sentence in the results section, “The scores on the Major Depression Inventory followed the trajectory of scores on self-reported health and showed a similar pattern of significant differences.” What does this mean for levels of depression among this sample?

Response: Table 3 shows similar patterns of significant differences between the moderate consumption trajectory and the remaining 5 trajectories for self-reported MDI, self-reported physical and mental health. The table also shows that the average score on the MDI was 5.47 with an SD of 6.34. Since the 10 items in the scale are scored on a 0 – 5 point scale, a mean of 5.47 is very low (Bech et al., 2001, report of a mean of 38.6 for 21 patients with major depression). 

Reviewer #2: I found the abstract well written and the sample size/scope impressive. I found the rationale in the introduction strong and easy to follow.

In the measures section, it was unclear to me whether the weekly number of units consumed over the entire life course was collected via timeline follow back or some other standardized tool. Perhaps it was similar to the cigarette pack years? Clarifying this would strengthen the paper. Moreover, it was unclear why the bottom age pa prioroeriod was 26-30 instead of 21-30. I assume this was a function of the way the questionnaire was programmed.

Response: We agree with the reviewer that this part of the methods section is too short and have revised paragraph:

“At the LiKO-15 follow-up, the participants completed a computer-based questionnaire, which included self-reported information on the weekly number of units over the entire life course, including the age periods 26–30 years, 31–40 years, 41–50 years, and 51–60 years. Data were also obtained for age-periods below age 26, but to focus on relatively stable adult drinking patterns, the analysis was restricted to the age-period 26-60 years. In Denmark, a standard drink or unit corresponds to 12 g of pure alcohol, roughly equivalent to 1 beer, 1 glass of wine or 4 cl of 40% liquor. The weekly number of units was calculated based on self-reported information on typical consumption of beer, wine, and liquor. Since the focus was on adult drinking trajectories, six a priori consumption trajectories were derived from the weekly number of units in the age periods 26–30 years, 31–40 years, 41–50 years, and 51–60 years. Men belonging to the first three stable trajectories were first identified, and then the remaining men were allocated to one of the other three trajectories” (revised manuscript page 6, line 129).

The LiKO-15 questionnaire also included the age periods below 15, 15-18, and 19-25. Thus, it was not possible to analyze 21-30 years, but we could have analyzed 19-30 years. However, adolescent alcohol consumption is likely to be more variable, and consequently we preferred to analyze the age period 26-60 years. This argument is expanded in the section paragraph of the introduction:

“Alcohol consumption habits develop during adolescence and young adulthood and are less

stable in this life period. This was confirmed in a meta-analysis finding dramatic shifts in 

alcohol intake in adolescence, but only moderate changes later in life [10], suggesting a shift from irregular drinking and binge-drinking in adolescence to a more stable drinking pattern later in adulthood. More recent studies have observed different trajectories of consumption in adolescence, including patterns of increasing and decreasing intake [11]. These findings and an observed peak around age 25 [8] suggest that a distinction should be made between adolescence and young adulthood where alcohol habits are being established and later adulthood where changes in consumption level and pattern can be observed but may be less frequent. Consequently, the present study focuses on adult trajectories of consumption after adolescence and before age-related changes in consumption after age 60 [9]”. (revised manuscript page. 4, line 81).

Reviewer #2: Finally, it was unclear why between 1 and 21 drinks per week was categorized as moderate consumption—this is a very large group with a wide range of consumption habits. More information here would be useful, world-count permitting. 

Response: An important background for the broad 1-21 drinks per week trajectory is the fact that Danish health authorities has recommended 21 drinks per week as an upper limit for sensible drinking for men since 1990 (there has been a recent change in these recommendations after our data were collected). Ignoring the few abstainers, the moderate consumption trajectory thus consists of men with a consumption level within recommended sensible limits. We were aware that 1-21 drink per week is a quite broad range, but a previous Danish study showed few differences between men drinking 1-9 drinks and 10-21 drinks per week (Mortensen et al., 2006). 

Reviewer #2: The rationale for transforming this measure to a binary measure of good vs. bad health was also unclear.

Response: This was mainly to simplify presentation, analysis and discussion of the results. The results in table 3 suggest that the binary measure was sensitive enough to show several significant differences among the trajectories.

Reviewer #2: In the data analysis section, it wasn’t stated why moderate consumption trajectory (through presumably because this was the largest group) was used as the reference, but otherwise the analyses seemed reasonable. 

Response: Since abstaining during adult life is relatively rare among Danish men, and not only this study, but also other Danish studies suggest that abstaining may be associated with less favorable characteristics (Mortensen et al., 2006), we did not use the abstaining trajectory as a reference. Among the remaining trajectories, the moderate and high-risk trajectories reflected a stable consumption level, but for several reasons, including the small cell size, it would be problematic to use the high-risk trajectory as reference. In contrast, the moderate trajectory included about two thirds of all men and had the most favorable characteristics on many variables. Consequently, it was used as reference. 

Reviewer #2: Some of the descriptive statistics in the beginning of the results are repetitive as these are already described in the prior sections. 

Response: In the revised manuscript we have tried to avoid unnecessary repetitions.

Reviewer #2: The discussion nicely highlights some key findings including that other studies “defining abstainers by self-report in late midlife will tend to mix the relatively few life-long abstainers with a larger number of former high-risk drinkers.”

My overall impressions of the manuscript are favorable.

Response: Thanks, for these positive comments.

 

References:

Bech P, Rasmussen NA, Raabæk Olsen L, Noerholm V, Abildgaard W. The sensitivity and specificity of the Major Depression Inventory using the Present Stata Examination as the Index of diagnostic validity. Journal of Affective Disorders, 66: 159-164, 2001.

Christensen GT; Molbo D; Angquist LH; Mortensen EL; Christensen K; Sørensen T I A. & Osler M. Cohort Profile: The Danish Conscription Database (DCD): A cohort of 728 160 men born from 1939 through 1959. International Journal of Epidemiology, 44 (2), 423-440, 2015.

Cohen J & Cohen P (1975), Applied multiple regression/correlation analysis for the behavioral sciences. Lawrence Erlbaum Associates, Hills, New Jersey,

Grønkjær M, Flensborg-Madsen T, Osler M, Sørensen HJ, Becker U, Mortensen EL

Adult-Life Alcohol Consumption and Age-Related Cognitive Decline from Early Adulthood to Late Midlife.Alcohol Alcohol. 2019a. pii: agz038. doi: 10.1093/alcalc/agz038.

Grønkjær M, Flensborg-Madsen T, Osler M, Sørensen HJ, Becker U, Mortensen EL.

Intelligence test scores before and after alcohol-related disorders - a longitudinal study of Danish male conscripts. Alcohol Clin Exp Res. 2019b. doi: 10.1111/acer.

Mortensen EL; Sørensen HJ; Jensen HH; Reinisch JM. & Mednick SA. IQ and mental disorder in young men.British Journal of Psychiatry, 187, 407-415, 2005.

Mortensen EL; Jensen HH; Sanders SA & Reinisch JM. .Associations between volume of alcohol consumption and social status, intelligence, and personality in a sample of young adult Danes. Scandinavian Journal of Psychology, 47, 387-398, 2006.

---

## [Decision Letter · Decision Letter 1]

8 Sep 2022

PONE-D-22-04437R1Social, psychological and health characteristics associated with stability and change in adult alcohol consumptionPLOS ONE

Dear Dr. Mortensen,

Thank you for submitting your manuscript to PLOS ONE. After careful consideration, we feel that it has merit but does not fully meet PLOS ONE’s publication criteria as it currently stands. Therefore, we invite you to submit a revised version of the manuscript that addresses the points raised during the review process.

We look forward to receiving your revised manuscript.

Kind regards,

Marc N. Potenza

Academic Editor

PLOS ONE

Journal Requirements:

Please review your reference list to ensure that it is complete and correct. If you have cited papers that have been retracted, please include the rationale for doing so in the manuscript text, or remove these references and replace them with relevant current references. Any changes to the reference list should be mentioned in the rebuttal letter that accompanies your revised manuscript. If you need to cite a retracted article, indicate the article’s retracted status in the References list and also include a citation and full reference for the retraction notice

Reviewers' comments:

Reviewer's Responses to Questions

**Comments to the Author**

1. If the authors have adequately addressed your comments raised in a previous round of review and you feel that this manuscript is now acceptable for publication, you may indicate that here to bypass the “Comments to the Author” section, enter your conflict of interest statement in the “Confidential to Editor” section, and submit your "Accept" recommendation.

Reviewer #2: (No Response)

2. Is the manuscript technically sound, and do the data support the conclusions?

Reviewer #2: Yes

3. Has the statistical analysis been performed appropriately and rigorously? 

Reviewer #2: Yes

4. Have the authors made all data underlying the findings in their manuscript fully available?

Reviewer #2: (No Response)

5. Is the manuscript presented in an intelligible fashion and written in standard English?

Reviewer #2: Yes

6. Review Comments to the Author

Reviewer #2: Thank you for the opportunity to re-review this manuscript.

I found the authors responses to the requirements of the journal adequate. I appreciate the authors’ response to the issue of studying sex differences raised by Reviewer 1. With regard to the lack of hypotheses, can the authors briefly state somewhere in the manuscript that these analyses were exploratory? I agree with R1 on point 13. The sentence “…participants who were abstaining from alcohol had the lowest intelligence score…” could be reworded to just state “…participants who were abstaining from alcohol had an intelligence score of M = 95.81…” to avoid any confusion. While the readership of PLOS One might not misinterpret this sentence, I think erring on the side of caution can’t hurt. This is a sensitive topic and if there is a chance that readers could infer that abstainers have an IQ below the normal range (even if the authors do not imply this direct relationship), that could be problematic.

I found the response to my concern about the methods section appropriate, especially the reference to the meta-analyses of alcohol intake in adolescents. I also found the response to my concern about the 1-21 drinks per week categorization acceptable given Danish standards for sensible drinking and few differences between subgroups who drink 1-9 vs. 10-21. In general, all of my concerns were addressed. A minor note: In response to R1’s request for more info on exclusion criteria, the authors mention “mental retardation” which I believe is now called “intellectual disability.”

7. PLOS authors have the option to publish the peer review history of their article (what does this mean?). If published, this will include your full peer review and any attached files.

Reviewer #2: No

---

## [Author Response · Author response to Decision Letter 1]

13 Sep 2022

PONE-D-22-04437R1

Social, psychological and health characteristics associated with stability and change in adult alcohol consumption

PLOS ONE

Author response to journal requirements and reviews

Journal Requirements:

To our knowledge we have not cited retracted papers

Reviewer 2:

I found the authors responses to the requirements of the journal adequate. I appreciate the authors’ response to the issue of studying sex differences raised by Reviewer 1. 

Response: we appreciate the reviewer’s positive evaluation of our response to the previous reviews.

With regard to the lack of hypotheses, can the authors briefly state somewhere in the manuscript that these analyses were exploratory? 

Response: Thank you for this suggestion. We have stressed the explorative nature of the study in both the introduction and the conclusion.

Line 106: The aims of this explorative study are to describe the prevalence of different adult-life alcohol consumption trajectories among Danish men and to analyze social, psychological, lifestyle and health characteristics associated with these adult-life consumption trajectories.

Line 539: The findings of this explorative study suggest that the majority of Danish men drink moderately in the life period from young adulthood to late midlife, and in a statistical sense stable moderate consumption is “normal” in Denmark.

I agree with R1 on point 13. The sentence “…participants who were abstaining from alcohol had the lowest intelligence score…” could be reworded to just state “…participants who were abstaining from alcohol had an intelligence score of M = 95.81…” to avoid any confusion. While the readership of PLOS One might not misinterpret this sentence, I think erring on the side of caution can’t hurt. This is a sensitive topic and if there is a chance that readers could infer that abstainers have an IQ below the normal range (even if the authors do not imply this direct relationship), that could be problematic. 

Response: We have followed the reviewer’s suggestion in the revised manuscript:

Line 297: Table 2 shows young adult characteristics in relation to consumption trajectories. The highest intelligence score and educational level were observed among moderate drinkers, while participants who were abstaining from alcohol had a mean intelligence score of 95.81.

I found the response to my concern about the methods section appropriate, especially the reference to the meta-analyses of alcohol intake in adolescents. I also found the response to my concern about the 1-21 drinks per week categorization acceptable given Danish standards for sensible drinking and few differences between subgroups who drink 1-9 vs. 10-21. In general, all of my concerns were addressed. 

Response: Thanks again for the positive comments.

A minor note: In response to R1’s request for more info on exclusion criteria, the authors mention “mental retardation” which I believe is now called “intellectual disability.”

Response: This is correct and line 156 has been revised accordingly: All Danish men – except those with disqualifying disease such as intellectual disability, diabetes, and epilepsy – must appear before the draft board at age 18 or seek permission to appear later.

---

## [Editor Report · Decision Letter 2]

31 Oct 2022

Social, psychological and health characteristics associated with stability and change in adult alcohol consumption

PONE-D-22-04437R2

Dear Dr. Mortensen,

We’re pleased to inform you that your manuscript has been judged scientifically suitable for publication and will be formally accepted for publication once it meets all outstanding technical requirements.

Kind regards,

Marc N. Potenza

Academic Editor

PLOS ONE
---

## [Editor Report · Acceptance letter]

3 Nov 2022

PONE-D-22-04437R2 

Social, psychological and health characteristics associated with stability and change in adult alcohol consumption 

Dear Dr. Mortensen:

I'm pleased to inform you that your manuscript has been deemed suitable for publication in PLOS ONE. Congratulations! Your manuscript is now with our production department. 

Kind regards, 

on behalf of

Dr. Marc N. Potenza 

Academic Editor

PLOS ONE